# Assessment of EU Bio-Based Economy Sectors Based on Environmental, Socioeconomic, and Technical Indicators

Víctor Fernández Ocamica *, Monique Bernardes Figueirêdo, Sebastián Zapata and Carmen Bartolomé

Research Centre for Energy Resources and Consumption CIRCE, 50018 Zaragoza, Spain; mbernardes@fcirce.es (M.B.F.); szapata@fcirce.es (S.Z.); cbartolome@fcirce.es (C.B.)
* Correspondence: vfernandez@fcirce.es

**Abstract:** The development of a resilient and circular bio-based economy is of paramount importance, notably in the EU, where current climate policies and evolving regulations strongly demand more sustainable practices, impacting monitoring and reporting, as well as the deployment of novel valorization routes for byproducts and waste streams. In this context, with the aim of assessing the current state of the European bio-based economy, a comprehensive analysis based on socio-environmental, socioeconomic, and technical indicators was carried out on major sectors, namely textiles, woodworking, pulp and paper, bio-based chemicals and materials, liquid biofuels, and bio-based electricity. Each sector was evaluated with respect to its main biological raw materials, and a methodology is proposed to link their geographical origin (inside or outside the EU), import shares, and internal production with socio-environmental impacts, based on official databases and indexes. Socioeconomic data (turnover and employment) and technical data (average bio-based content within the main products of the sector) were also considered for the analyses, allowing a multi-angle comparison between sectors and the identification of barriers and opportunities for future developments. Finally, a quantitative and qualitative overview of non-hazardous biogenic waste streams generated in the EU is presented, and opportunities for their valorization and reintegration into the EU bio-based economy are discussed. As a result of this analysis, beyond enabling the assessment of each sector within the bio-based economy, along with the assignment of values for comparison, the implementation of this evaluation facilitated the identification of improvement pathways, which were consolidated into a set of proposals.

**Keywords:** raw materials; environmental indicators; socioeconomic indicators; waste valorization; bio-based products

## 1. Introduction

There has been a growing and widespread interest, whether from a social, business, or legislative perspective, in achieving sustainability. This involves not only modernizing the sectors that make up the economy, notably the industrial sectors, but also strengthening the EU's position in the global economy and ensuring the prosperity of its citizens. To achieve these goals, among other things, innovative approaches have been sought to produce the products and materials used by all of us. One of these approaches is further developing the bioeconomy, which encompasses the production of renewable biological resources, including waste streams and byproducts, and their conversion into value-added products [1]. Since 2012, when the first Bioeconomy Strategy was published, the Bioeconomy Strategy has been considered an important component in the implementation of the European Green Pact, thanks to the contributions it has made over the years and will continue to generate. Such relevance has been maintained over time, leading to an update of the Bioeconomy Strategy in 2018 and the development of a progress report on the implementation of the Bioeconomy Strategy for its tenth anniversary (2022).

The importance of the bioeconomy is not limited to sustainability. In fact, the bioeconomy is responsible for generating a turnover value of EUR 2.3 trillion (2020), creating

jobs, especially in coastal and rural areas, thanks to the growing participation of primary producers in important value chains, building a future in line with the climate targets of the Paris Agreement, creating new value chains and greener and more cost-effective industrial processes, turning biowaste, residues and discards into valuable resources, and creating innovations and incentives to help retailers and consumers to reduce waste generation [2].

Accordingly, the bioeconomy encompasses and intertwines terrestrial and marine ecosystems, along with the services they provide, all primary production sectors utilizing and generating biological resources (agriculture, forestry, fisheries, and aquaculture), and all economic and industrial sectors that harness biological resources and processes to produce food, feed, bio-based products, energy, and services [2]. The industrial sectors that rely on biological resources consist of six key areas: (i) the textiles and textile products sector; (ii) the wood products and furniture sector; (iii) the paper sector; (iv) the bio-based chemicals, pharmaceuticals, rubber, and plastics sector; (v) the biofuels sector; and (vi) the bio-based electricity sector.

This article presents the results of a series of analyses aimed at assessing the status of each of these sectors in comparison to one another with the goal of proposing solutions and prioritizing aspects to foster their development from a combined social, environmental, economic, and technical perspective. To achieve this, as depicted in Figure 1 and described in the methodology section, three approaches were taken. Firstly, a set of known socio-environmental parameters was employed to evaluate the sectors based on the key bio-based raw materials they currently utilize. Secondly, these same sectors were analyzed in terms of socioeconomic data, taking into consideration the turnover, employment levels, and "bio-based product output share", resulting in the real contributions of the bio-based portion within each of these sectors. Lastly, a review of the literature was conducted to compile potential applications of biological-origin waste as raw materials within each sector. Overall challenges and opportunities are discussed, and recommendations to propel the EU bio-based economy are given based on the assessment's results.

**Figure 1.** Overview of the main EU bio-based sectors assessed in this study, their main raw materials and products, and the main biogenic waste streams generated by the EU.

This article is structured as follows: Following the introduction, the methodology section details the main databases used and the calculations defined to obtain socio-environmental scores per raw material and per sector. Section 3 compiles the results from the analyses related to both the socio-environmental and socioeconomic parameters. Section 4 describes the EU panorama of non-hazardous biogenic waste generation (volumes and origins) and presents the results of the potential use of said streams as raw materials in the addressed bio-based economy sectors. Section 5 presents a discussion of the results

obtained and compiles the proposals for improvement based on these results. Section 6 presents a set of key conclusions obtained from the assessment, including recommendations related to data availability, uncertainty, and identified gaps that can be improved for future assessments.

## 2. Methodology

To conduct this assessment, the methodology employed can be delineated into two primary stages. The first stage focused on evaluating the main bio-based products and sectors to conduct an analysis of the current state of the situation of these sectors in the bio-based economy at the European Union level (Section 2.1), while the second stage scrutinized the potential utilization of biowaste as a raw material within the previous sectors that make up the bio-based economy (Section 2.2). Detailed descriptions of these stages are provided in their respective sections below.

In a broader context, as illustrated in Figure 2, the methodology commenced with an initial review of the literature and data. These data served as the foundation for a series of calculations, culminating in a value that facilitated the assessment and comparison of sectors, as well as the analysis of the biowaste that could be used as raw material.

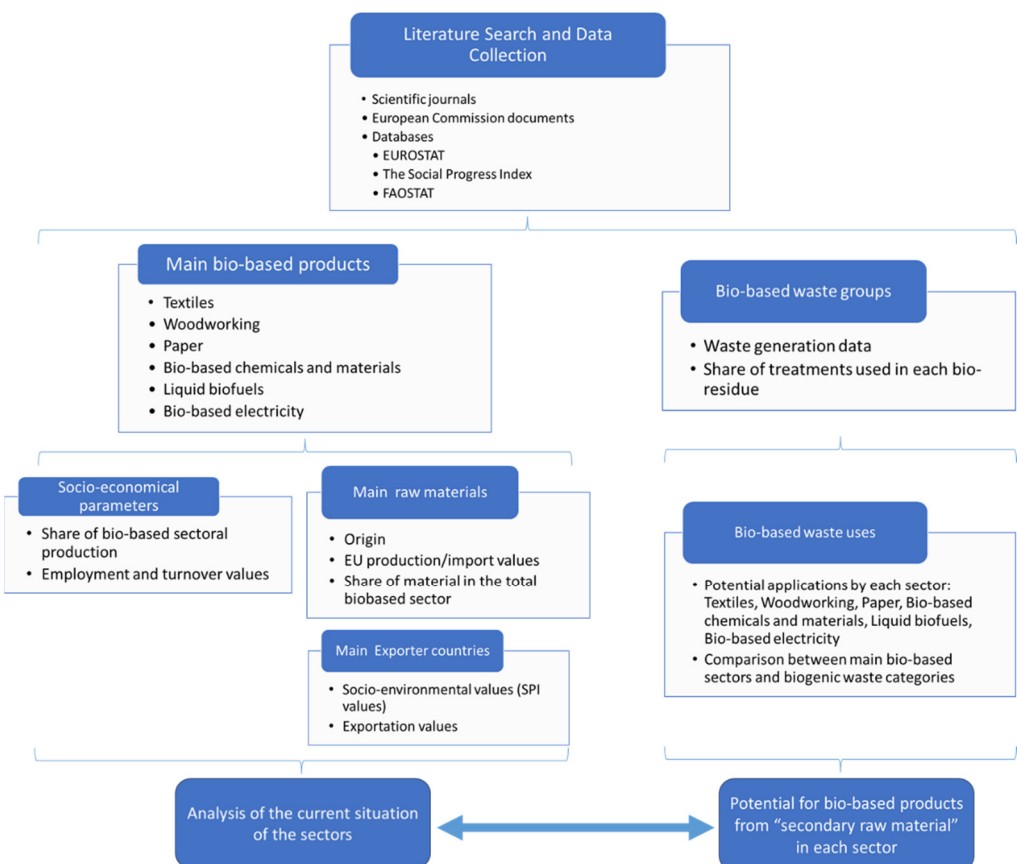

**Figure 2.** Schematic overview of the methodology employed.

### 2.1. Methodology for the Analysis of the Current Situation of the Main Bio-Based Economy Sectors

The methodology followed for the development of Section 3 of this article consists of two main phases. In the first phase, a literature search and data collection were conducted to describe the European bio-based economy and its constituents. Various reputable scientific databases and publishers were consulted for article retrieval, along with websites, sector association reports, European Commission documents, and relevant databases (EUROSTAT, the Social Progress Index (SPI), and the FAO (Food and Agriculture Organization)).

As an initial foundation for this literature search, we utilized the six main sectors comprising the bio-based economy. These sectors, along with similar keyword sets, were employed as search terms in scouring a variety of scientific, institutional, and sector-specific documents. By employing this method, we identified the three primary raw materials used in each of these bio-based sectors, along with their respective shares of the total material composition within the bio-based sector.

Once these raw materials were identified, the FAOSTAT database, among others, was consulted to determine the production/import ratio of each material in relation to its consumption. In cases where it was necessary, the three primary exporting countries of such materials to the EU were identified. When dealing with a material that is imported, either entirely or partially, into the EU, in addition to identifying the top three countries exporting that material to the EU, the respective quantities from each country were compiled. This process aimed to determine the influence of each country on the importation of the specific material. These countries, along with the EU, became the focus of the search for the value assigned (SPI Overall Scoring in [3]) via the Social Progress Index (SPI). In addition to this, utilizing available databases from the EU, values for each sector, including employment and turnover, were collected. Simultaneously, consideration was given to the share of bio-based sectoral production.

In the research process for this article, multiple sources of information were used for data collection and methodology development; however, a number of these sources, compiled in Table 1, are of particular importance.

**Table 1.** Sources used for the analysis of the current situation of the main bio-based sectors.

| Title | Source Type | Application | Year | Source |
|---|---|---|---|---|
| The Role of Bio-Based Textile Fibres in a Circular and Sustainable Textiles System | Report | Textile sector | 2023 | [4] |
| Preferred Fiber & Materials | Report | Textile sector | 2022 | [5] |
| Key Statistic 2022, European Pulp & Paper Industry | Booklet | Pulp and paper | 2023 | [6] |
| Insights into the European Market for Bio-Based Chemicals | Report | Chemical sector | 2019 | [7] |
| European Bioeconomy in Figures 2008–2019 | Report | Chemical sector | 2022 | [8] |
| European Union: Biofuels Annual | Report | Liquid biofuels sector | 2022 | [9] |
| FAOSTAT | Database | Production and trade, all sectors | - | [10] |
| Social Progress Index | Database | Social and environmental situation, all sectors | 2022 | [3] |
| Jobs and Wealth in the European Union Bioeconomy | Database | Employment, turnover, and bio-based share, all sectors | 2020 | [11] |

In the second phase, a series of parameters was established to characterize the sectors and enable their comparison. Depending on the parameters and type of information available for each sector, the data underwent a standardization process to ensure comparability. The aspects addressed in this study include the following:

- The share of bio-based sectoral production, employment, and turnover values.

Data published on the European Commission's Resource Economy Data Modelling Platform was used to analyze the bio-based shares, the employment, and the turnover values in the studied sectors.

- Raw materials in the bio-based value chains and their origins.

Through an extensive literature search, the main raw materials, as well as their shares, used in the production of the leading bio-based products in each sector were identified. Subsequently, using the FAO database, the production, import, and export values of these raw materials were determined, excluding data related to intra-European Union trade.

- Effects from a social and environmental point of view.

Based on this information, the ranking established by the Social Progress Index (hereafter referred to as SPI) was consulted; it evaluates the social and environmental situation of 169 countries with all its indicators and 27 others with part of them.

For this purpose, the index uses a set of 60 indicators, exclusively social and environmental. These indicators are distributed over 12 groups, which, as can be seen in Figure 3, can in turn be distributed into three main frameworks, namely basic human needs, foundations of wellbeing, and opportunity.

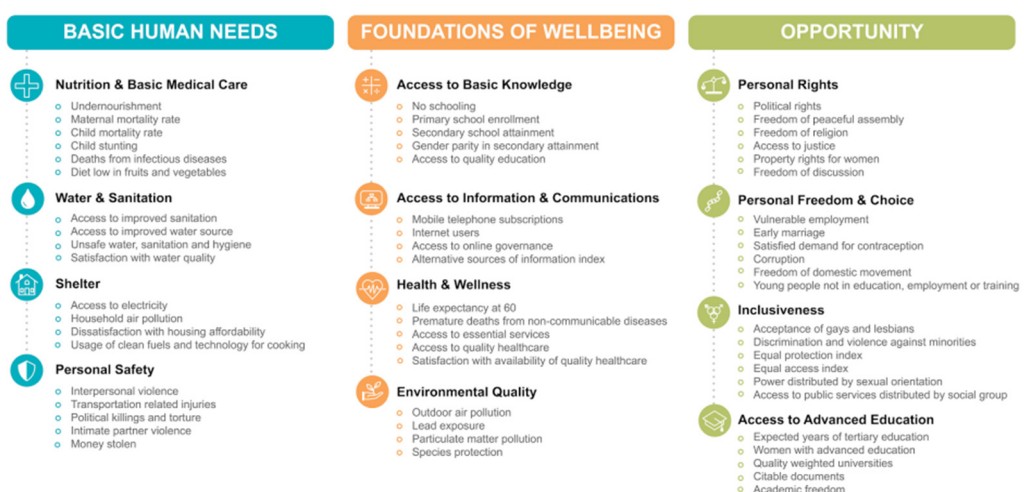

**Figure 3.** 2022 Social Progress Index® Framework [3].

Using these assessments, the average SPI value for the European Union (EU; 85.29) was calculated and then multiplied by the percentage of the corresponding raw material produced within the EU in relation to the total quantity available in the EU (see Equation (1)).

$$PV_m = EUv \times P\%_m \qquad (1)$$

where the following applies:

$PV_m$ is the value of the SPI, corresponding to the analyzed material, m, that has been produced exclusively within the EU.

$EU_V$ is the average SPI value of the 27 EU members.

P% is the percentage of production, within the EU, of the material m being analyzed in relation to the total available, which is the sum of the tonnes produced of that material and the tonnes imported into the EU (excluding internal trade).

Furthermore, considering the export values and the SPI of the three main countries that export to the EU, the situation of each identified raw material was evaluated. For this purpose, the exported tonnage from each country was aggregated, and the corresponding percentage of each country with respect to this sum was multiplied by the SPI result. Then, the three values were summed, resulting in a unique value (see Equation (2)).

$$IV_m = \sum_{x1}^{x3} \left( \frac{Ex, m}{\sum Ex, m} \times EVx \right) \times I\%m \qquad (2)$$

where the following applies:

$IV_m$ is the value of the social index of the analyzed material, m, which has been consumed in the EU and which is imported.
$E_{x,m}$ is the quantity of the analyzed material, m, exported to the EU by one of the three main exporters, x.
$EV_x$ is the value in the SPI of the exporting country, x.
$I\%_m$ is the percentage of importation of the material, m, which is being analyzed in relation to the total available (imports + internal production).

The importation value was added to the production value to give a unique Consumption value per raw material (see Equation (3)).

$$CV_m = IV_m + PV_m \tag{3}$$

where the following applies:

$CV_m$ is the value of the SPI of the analyzed raw material, m, consumed in the EU.

The aim was to obtain a value reflecting the social and environmental impacts of consuming each selected raw material in the EU. Finally, after identifying the values for each raw material, they were multiplied by the corresponding percentage that represents the use of that raw material in the sector. The results were then summed, yielding a value that represents the environmental and social impact of each sector (see Equation (4)).

$$SV_s = \sum_{m1}^{m3}(CV_m \times MS\%_m) \tag{4}$$

where the following applies:

$SV_s$ is the value of the SPI of the sector, s, under analysis.
$MS\%_m$ is the share of material m in the total bio-based sector studied, represented by the three main bio-based materials.

These results retain the characteristics of the SPI scoring system, so the lower the score, the worse it is in terms of social and environmental impact for the analyzed sector.

The results of all of these data compilations and operations are presented in two tables for each sector throughout Section 3. The first of these tables, unless there are sector-specific peculiarities, as described in the respective subsection, highlights the gathered data for the top three raw materials in each sector. For each material, it includes the three main exporting countries, intermediate operations, and the final outcome, characterizing the sector from a socio-environmental perspective. Meanwhile, the second table outlines all data related to socioeconomic parameters.

*2.2. Methodology for Assessing the Potential for Bio-Based Products from Secondary Raw Materials*

Like the previous section, this part of the methodology began with a literature and data search, focusing in this case on non-hazardous biowaste groups listed in Eurostat's waste generation database. From the database itself, the corresponding values for waste generation and the share of treatments used in each bio-residue were extracted. Meanwhile, the waste groups themselves, along with their constituents and the bio-based economy sectors, served as search elements to identify cases/studies in which these residues had been utilized or demonstrated to be potential raw materials for products within the sectors under analysis.

As a result of this search exercise, a total of 60 references were compiled, exemplifying the utilization of one or multiple residues from the studied waste groups as raw materials in the production of products within the sectors constituting the bio-based economy. Using the list of cases generated from this search, a comparison was created between the waste groups, based on the applications they received, and the studied sectors. This was done to estimate the capacity of each sector to employ each waste group as a raw material.

As in Section 3, Section 4 consisted of two phases. It began by presenting the existing situation and major contributors to non-hazardous biowastes, and then it proceeded to explore potential applications of these wastes in the context of the bio-based economy. For this purpose, an extensive bibliographic search was conducted. Finally, the different raw materials demanded by the sectors were cross-referenced to detect potential competition between them. In this way, in addition to identifying the potential applications of biowastes, this section provides recommendations regarding the selection of biowastes to reduce likely future supply problems.

## 3. Analysis of the Current Situation of the Sectors That Make up the Bio-Based Economy

### 3.1. Sector: Textiles and Textile Products

The textile sector, whether to produce fabrics, clothing, technical textiles, or various household products, is based on textile fibers. This industry grew significantly, almost doubling in value from 58 million tonnes in 2000 to 113 million tonnes in 2021. Moreover, this positive trend is expected to continue, with a projected growth of approximately 149 million tonnes by 2030 [5].

Global fiber production has been dominated since 1990 by synthetic fibers, of which around 72.2 million tonnes (64% of the fibers produced in 2021) were mostly polyesters (60.5 million tonnes). In addition to these, plant fibers also have a large market share, accounting for 28% of the total, i.e., around 31.3 million tonnes, of which 24.7 million tonnes were cotton. The rest of the market is made up of fibers of animal origin (wool, silk, etc.), accounting for 1.6%, and man-made cellulosic fibers with 6.4% [5].

In the context of textiles, the term "bio-based" refers to the origin of the carbon structure of the polymer fiber and whether this origin is from a renewable source. In other words, it refers to whether the fiber used in the textile is composed of materials derived from natural and renewable sources, as opposed to synthetic materials derived from non-renewable resources such as petroleum. For example, only 0.02% of all polyester produced and 0.4% of all polyamides produced are considered bio-based. This is why the current manufacture of textiles and textile products of organic origin is mainly based on cotton fibers, followed with a significant difference by viscose derived from wood pulp and jute. In the case of cotton and wood pulp, the demand for both raw materials is covered mainly by the EU's own production, at around 80% for both materials, while in the case of jute, it comes entirely from outside the EU [5,10].

Table 2 displays, for each of the three main raw materials utilized in the sector, the key countries exporting each of these raw materials to the EU. It also includes the value received by each country in the SPI and the resultant averaged value from the combination of the three sources. This average calculation considers the quantity exported by each country, as well as the distribution between importation and production. Additionally, intermediate calculations are considered in determining the sector's SV (sector value); see Section 2 for details.

As shown in Table 2, the bio-based fraction of the textile sector, according to the origin of its main raw materials, their SPI value, and the share of each raw material in the sector (MS%), has an estimated SV of 80.66 [7].

In addition to the impacts associated with the sector based on the origin of its raw materials, it is noteworthy that, among all bio-based fibers, cotton has the most significant impact on factors such as land and water use, eutrophication, and toxicity, being surpassed in terms of climate change only by viscose, which also generates a notable concern in terms of toxicity and land use [4].

**Table 2.** Textile sector—SV calculation.

| Raw Material | Top Three Non-EU Exporters [10] | Export Share [10] | EV$_x$ [1] [3] | Averaged SPI per Raw Material | I%/P% [2] | IV$_m$ [3] | PV$_m$ [3] | CV$_m$ [4] | MS% [5] | SV$_s$ [6] |
|---|---|---|---|---|---|---|---|---|---|---|
| Cotton | USA | 47% | 84.65 | | | | | | | |
| | Brazil | 32% | 71.26 | 75.33 | 21%/79% | 15.76 | 67.45 | 83.21 | 73% | |
| | India | 21% | 60.19 | | | | | | | |
| Wood pulp | Brazil | 55% | 71.26 | | | | | | | 80.66 |
| | USA | 29% | 84.65 | 76.62 | 20%/80% | 15.32 | 68.23 | 83.56 | 17% | |
| | Uruguay | 16% | 80.27 | | | | | | | |
| Jute | Bangladesh | 78% | 56.06 | | | | | | | |
| | India | 16% | 60.19 | 56.65 | 100%/0% | 56.65 | - | 56.65 | 10% | |
| | URT | 6% | 54.87 | | | | | | | |

[1] SPI value of the exporting country. [2] Importation and production percentage, within the EU, of the raw material being analyzed in relation to its sum. [3] Social index value of the imported/produced fraction of the analyzed raw material, m, consumed in the EU. [4] SPI value of the analyzed raw material, m, resulting from the sum of IV + PV. [5] Proportion of raw material, m, in the total surveyed sector represented by the three main bio-based raw materials. [6] SPI value of the sector under analysis.

As for the final products that make up the bio-based fraction of the sector, the distribution used by the NACE (Nomenclature of Economic Activities) code was used, resulting in three main products: textiles (C13), wearing and apparel (C14), and leather products (C15) [11,12]. As shown in Table 3, these products have an average of 42% bio-based content and contribute to the generation of more than 700,000 jobs and a turnover of 72,000 million euros [11]. At the same time, it can be noted that, while each product type contributes similarly to the overall turnover among the three, there is a noticeable difference in the number of employees required to achieve these values, particularly in the case of wearing apparel. In this case, the production requires more than 100,000 extra-employed people to reach the turnover levels of the other two products. This suggests a relatively low value of the "wearing apparel" product category compared to the others.

**Table 3.** Socioeconomic data of the bio-based fraction of the textile sector. Data source: [11].

| | Employment, 2020 | Turnover (Million EUR), 2020 | Bio-Based Output Share, 2020 | Turnover (EUR) per Person Employed |
|---|---|---|---|---|
| Bio-based textiles | 210,968.49 | 26,370.43 | 39.20% | 124,997.01 |
| Bio-based wearing apparel | 318,122.40 | 23,845.14 | 39.80% | 74,955.87 |
| Leather | 194,722.99 | 21,870.28 | 50.40% | 112,314.83 |
| Total | 723,813.88 (sum) | 72,085.85 (sum) | 42% (average) | 99,591.69 (average) |

### 3.2. Sector: Wood Products and Furniture

The EU forestry industry comprises four main sectors, namely woodworking, furniture, pulp and paper manufacturing and converting, and printing [13]. The most important sub-sectors of the forest-based industry in Europe are wood and furniture, consisting of sawmilling (15%), construction wood products (37%), and furniture manufacturing (48%), which use up to 70% of the wood consumed in Europe [14,15].

Apart from the economic and employment benefits discussed below, this sector is credited with several other positive contributions, including the following [14,15]:

- Supporting rural areas by helping maintain employment and generating wealth.
- Contributing to the goal of a low-carbon bioeconomy through the carbon storage properties of wood and its fractions.
- Promoting sustainability through reusability and recyclability; for example, sawmill by-products like shavings and sawdust are transformed into wood-based panels.
- Enabling the repurposing of wood-based materials as energy sources at the end of their useful lives.

Similar to the textile sector, the European wood sector covers most of the internal wood demand except for tropical wood, which is 100% imported. Nonetheless, the tropical wood volumes used are much lower than those of hardwood and softwood [10]. Table 4 presents data on the primary raw materials and their significance within the sector. It includes information on the main exporters of these materials to the EU, the SPI, the distribution of exports from these countries, and the breakdown between importation and production. Additionally, it outlines the calculations used to assign a sector value (SV) to the sector.

In the case of this sector, which is mostly characterized by softwood lumber, a SV of 85.17 is estimated, as shown in Table 4. It is very similar to the EU SPI average (85.29). However, the sector still faces the following challenges [14,16]:

- Supply problems due to unaffordable prices resulting from competition with the (often subsidized) bioenergy industry and countries with low production costs.
- A labor shortage as a consequence of the aging workforce and the reluctance of young people to enter the sector.
- The widespread use of formaldehyde (classified as a carcinogenic compound)-based adhesives on wood panels such as fiberboards and plywood.

**Table 4.** Wood products and furniture—SV calculation.

| Raw Material | Top Three Non-EU Exporters [10] | Export Share [10] | $EV_x$ [3] | Averaged SPI per Raw Material | I%/P% | $IV_m$ | $PV_m$ | $CV_m$ | MS% | $SV_s$ |
|---|---|---|---|---|---|---|---|---|---|---|
| Hardwood | Russia | 74% | 71.99 | 73.69 | 10%/90% | 7.26 | 76.89 | 84.15 | 21% | |
| | USA | 14% | 84.65 | | | | | | | |
| | Belarus | 12% | 71.49 | | | | | | | |
| Tropical hardwood | Papua New Guinea | 39% | 48.12 | 56.79 | 100%/0% | 56.79 | - | 56.79 | 0% | 85.17 |
| | Brazil | 32% | 71.26 | | | | | | | |
| | Solomon Islands | 29% | 52.40 | | | | | | | |
| Softwood | Norway | 55% | 90.74 | 82.29 | 3%/97% | 2.73 | 82.73 | 85.46 | 79% | |
| | Russia | 27% | 71.99 | | | | | | | |
| | Belarus | 18% | 71.49 | | | | | | | |

As with the previous sector, two main product groups were defined from the list of NACE codes. Wood products correspond to C16 and are regarded, after excluding components such as paints or adhesives, as 100% bio-based, while furniture corresponds to C31, and the estimate for bio-based (wood) content in furniture is 45.60% [11,12]. Table 5 shows that these two product categories account for more than 1,300,000 jobs and over 170,000 euros in turnover. These values, respectively, represent 41% and 25% of the total employment and turnover generated within the bio-based economy, and will be further discussed in Section 5.

**Table 5.** Socioeconomic data of the bio-based fraction of the wood products sector. Data source: [11].

| | Employment, 2020 | Turnover (Million EUR), 2020 | Bio-Based Output Share, 2020 | Turnover (EUR) per Person Employed |
|---|---|---|---|---|
| Wood products | 907,712.22 | 131,359.05 | 100% | 144,714.42 |
| Wooden furniture | 419,396.72 | 42,132.23 | 45.60% | 100,459.13 |
| Total | 1,327,108.94 (sum) | 173,491.28 (sum) | 72.40% (average) | 130,728.74 (average) |

*3.3. Paper and Paper Products*

Another crucial sub-sector of the forestry industry within the bio-based economy is the pulp and paper industry. Throughout its history, this industry has been renowned for its contributions to the bioeconomy, extending beyond aspects related to job generation and turnover. Noteworthy achievements include the following [17]:

- Energy self-sufficiency.
- A significant reduction in net $CO_2$ emissions by generating over half of its primary energy from biomass.
- A paper recycling rate exceeding 70%.

In line with the Confederation of European Paper Industries (CEPI), which encompasses 16 out of the 27 EU members and is, therefore, sufficiently representative to indicate production/import/export ratios for the EU, a total of 72.5 million tons of paper and cardboard were consumed (produced + imported—exported) within the CEPI member countries in 2022. These products were sourced from raw materials such as pulp (wood and other fibers), non-fibrous materials, and recycled paper. As for pulp, it can be produced through the consumption of wood, which is the main route, and other fibers (straw, bamboo, bagasse, etc.). In addition, pulp can be categorized into two types: integrated pulp, produced by the paper mills themselves, and market pulp, which is produced by third-party entities. The consumption of each raw material is shown in Table 6 [6].

**Table 6.** European pulp and paper industry—raw materials overview (million tonnes). Data source: [6].

| Raw Material | Consumption | Production | Imports | Exports |
|---|---|---|---|---|
| Integrated pulp | 20.55 | 20.55 | - | - |
| Market pulp | 16.86 | 14.70 | 7.48 | 5.32 |
| Paper for recycling | 47.49 | 52.63 | 2.01 | 7.15 |
| Non-fibrous materials | 11.74 | 11.74 | - | - |
| Total | 96.63 | 99.61 | 9.50 | 12.48 |

As mentioned above, commercial pulp is made for the most part from wood (99.88%). However, not all this wood originates in the EU. Accordingly, 10.4% (6.4% roundwood and 4% wood chips) of the wood consumed by the CEPI members is imported, from which 6.2% is imported from EU countries. It was, therefore, assumed that the EU consumption of internal wood for pulp production is 95.8%, while the remaining 4.2% of the wood used for pulp production is imported.

Furthermore, within the two types of pulp, several sources can be distinguished. In the case of market pulp, there is pulp produced within the EU from internal wood (63%), pulp

produced within the EU from imported wood (3%), and imported market pulp (34%). In the case of integrated pulp, a distinction is made between pulp produced from internal wood (95.8%) and imported wood (4.2%). As a result, and as shown in Table 7, the raw materials imported for paper production are paper for recycling, imported market pulp, and wood used for pulp production. In addition to not assessing the impact of non-fibrous materials, due to being considered negligible and lacking the same quality of data as other materials, it is important to note that, as a result of data availability, the origins of the remaining materials have often been limited to regions rather than specific countries. In addition, to be able to carry out the calculations in Table 7, the paper production through each source has been estimated. For this purpose, the following conversion factors have been used: to produce 1 kg of paper, 1.09 kg of pulp or 1.34 kg of wastepaper is needed [6,18–20].

**Table 7.** Pulp and paper—SV calculation.

| Raw Material | | Top Three Non-EU Exporters [10] | Export Share [10] | $EV_x$ [3] | Averaged SPI per Raw Material | I%/P% | | $IV_m$ | $PV_m$ | $CV_m$ | MS% | $SV_s$ |
|---|---|---|---|---|---|---|---|---|---|---|---|---|
| Paper for recycling | | Other Europe North America Asia | 85% 15% 1% | 80.23 85.01 66.33 | 80.87 | 4%/96% | | 2.96 | 82.17 | 85.13 | 51% | |
| Pulp | Imported Pulp | Latin America North America Other Europe | 77% 16% 7% | 69 85.01 80.23 | 72.31 | 21% | 79% | 15.38 | 67.38 | 82.77 | 49% | 83.97 |
| | Pulp from imported wood | Other Europe Brazil Russia | 77% 12% 11% | 80.23 71.26 71.99 | 78.21 | | | | | | | |

In the case of the EU paper sector, according to the established methodology, an SV of 83.97 was estimated. However, as in the case of wood, a high SV does not preclude the existence of multiple challenges that the sector has to face, such as the following [17]:

- High consumption of raw materials and energy, with high capital costs and long investment cycles.
- A decline in the consumption of some paper products as a result of digitization.
- An increase in exports, but tariff barriers and protectionist subsidies for rival goods create an uneven playing field.
- Supply issues arising from the growing demand for bioenergy companies.
- Economic disadvantages compared to other competitors due to rising energy prices.

As can be seen in Table 8, the socioeconomic data of the bio-based fraction of the paper sector (data source: [11]) consists of only one category or NACE Code (C17). Nonetheless, this category is of great importance, especially in terms of turnover, as it represents 26% of the turnover of the bio-based economy.

**Table 8.** Socioeconomic data of the bio-based fraction of the paper sector. Data Source: [11].

| | Employment, 2020 | Turnover (Million EUR), 2020 | Bio-Based Output Share, 2020 | Turnover (EUR) per Person Employed |
|---|---|---|---|---|
| Total | 616,636.50 (sum) | 177,034.35 (sum) | 99.50% (average) | 287,096.77 (average) |

### 3.4. Bio-Based Chemicals, Pharmaceuticals, Rubber, and Plastics

In general terms, products within this sector (excluding biofuels, which will be discussed separately in Section 3.5) are composed of organic material (from fossil and bio-based sources), which is the only component that can be made fully bio-based, and inorganic material. That is why, despite offering a wide variety of fully bio-based products (e.g., natural dyes and pigments, enzymes, and fatty acids), the vast majority of bio-based products are only partially bio-based in this sector. Evidence of this is seen in the classification of

the 534 products under NACE division 20 ("manufacture of chemicals and chemical products"), of which 110 have attained bio-based status, indicating the presence of bio-based ingredients, even if in small quantities. Among these, only 44 products are exclusively derived from bio-based materials. The rest comprise 26 products with a minimum bio-based content of 10% and 40 products with less than 10% bio-based content [8].

As a result of the diversity in bio-based content depending on the type of product, the subsectors of this industry exhibit the following bio-based output shares, according to the value added by each of them, which refers to the gross income after adjusting for operating subsidies and indirect taxes [11]:

- Bio-based chemicals (excl. biofuels) (NACE Code 20)—8%.
- Bio-based pharmaceuticals (NACE Code 21)—49%.
- Rubber and bio-based plastics (NACE Code 22)—4%.

Due to variations in production volumes across each sector, these proportions translate into an average of 22% of the value generated by the above-mentioned products originating from bio-based materials [11].

Regarding the raw materials used in the bio-based value chains, the following ones stand out: sugar, starch, vegetable oil, and wood. Similarly, for the bio-based fraction, the proportions of the raw materials used vary according to the product and subsector. However, in general terms, it has been reported that the primary raw material is vegetable oil, accounting for 62% of the feedstock used, followed by sugar/starch at 19% and wood at 17% [7].

For materials using wood (wood pulp) and sugar/starch (corn, sugar crops, wheat, etc.) as raw materials, the EU demand would be internally covered by 100% and 90%, respectively. However, vegetable oils (palm oil, soybean, rapeseed, etc.), which, as highlighted before, constitute the most used raw material in the sector, show a high dependence on imports to the EU at 79% [7].

Table 9 shows the main data regarding the bio-based chemicals sector, based on the three main vegetable oils and the three main sources of sugar and starch in the sector.

In this case, unlike the last two sectors, the bio-based chemicals sector has an SV (78.18) that is farther away from the EU SPI average due to a higher dependency on importing raw materials from countries with a lower SPI. Along with this, one should consider all the aspects that threaten the sector, such as high production costs, lower performance in some cases when compared with fossil-based equivalents, regulatory barriers for new chemicals and materials, competition with food and/or energy supply chains, and a variety of shortcomings such as a lack of awareness, incentives, investment, and infrastructure.

It is important to mention the existing negative public opinions on palm oil and soybean oil, which, as can be seen in Table 9, are the most important in the sector [21]. These opinions highlight that, even though these sources can produce more oil per surface area than other types of oilseed plantations, indisputable effects are generated by their production, the most prominent being deforestation, drainage, and peatland burning in South Asian and American countries [22–24], where SPI values are considerably lower than in the EU. Other issues related to these value chains involve poor labor conditions and, in some cases, evidence of child labor [25].

In socioeconomic terms, the sector is responsible for generating significant contributions in terms of employment and turnover, as can be observed in Table 10.

**Table 9.** Bio-based chemicals sector—SV calculation.

| | Raw Material | Top Three Non-EU Exporters [10] | Export Share [10] | $EV_x$ [3] | Averaged SPI per Raw Material | I%/P% | $IV_m$ | $PV_m$ | $CV_m$ | MS% | $SV_s$ |
|---|---|---|---|---|---|---|---|---|---|---|---|
| Veg. oil | Palm oil | Indonesia<br>Malaysia<br>Guatemala | 64%<br>34%<br>2% | 66.67<br>74.08<br>60.21 | 69.06 | 100%/0% | 69.06 | - | 69.0 | 35% | 78.18 |
| | Soya bean oil | Argentina<br>Brazil<br>USA | 67%<br>23%<br>10% | 78.64<br>71.26<br>84.65 | 77.53 | 28%/72% | 21.71 | 61.41 | 83.12 | 31% | |
| | Rapeseed oil | Canada<br>Russia<br>Belarus | 73%<br>19%<br>8% | 88.17<br>71.99<br>71.49 | 83.83 | 24%/76% | 20.12 | 64.82 | 84.94 | 11% | |
| Sugar/starch | Maize | USA<br>Argentina<br>Ukraine | 53%<br>28%<br>19% | 84.65<br>78.64<br>74.17 | 81.01 | 35%/65% | 28.35 | 55.44 | 83.79 | 12% | |
| | Wheat | Canada<br>Russia<br>Australia | 63%<br>20%<br>17% | 88.17<br>71.99<br>87.83 | 84.96 | 20%/80% | 16.99 | 68.23 | 85.22 | 6% | |
| | Sugarcane | Brazil<br>Thailand<br>Australia | 82%<br>10%<br>8% | 71.26<br>69.8<br>87.83 | 72.41 | 100%/0% | 72.41 | - | 72.41 | 5% | |

**Table 10.** Socioeconomic data of the bio-based fraction of the bio-based chemical products sector. Data source: [6].

| | Employment, 2020 | Turnover (Million EUR), 2020 | Bio-Based Output Share, 2020 | Turnover (EUR) per Person Employed |
|---|---|---|---|---|
| Bio-based chemicals | 97,803.43 | 38,248.83 | 8% | 391,078.62 |
| Bio-based pharmaceuticals | 314,632.65 | 176,993.44 | 49% | 562,539.97 |
| Rubber and bio-based plastics | 60,510.42 | 10,864.27 | 4% | 179,543.79 |
| Total | 472,946.50 (sum) | 226,106.54 (sum) | 17% (average) | 391,078.62 (average) |

### 3.5. Liquid Biofuels

Biofuels are fuels derived directly or indirectly from biomass, and they can be differentiated into three types [26]:

- Solid biofuels (fuelwood, wood residues, wood pellets, animal waste, vegetal material, etc.): this category comprises solid, non-fossil organic matter of biological origin (e.g., biomass) that can be used as fuel for heat or electricity production.
- Liquid biofuels (bio-gasoline, biodiesel, bio-jet kerosene, etc.): this group includes all liquid fuels produced from biomass and/or the biodegradable fraction of waste. These are suitable for blending with fossil-derived liquid fuels or replacing them.
- Biogases (from anaerobic fermentation and thermal processes): These gases are primarily composed of methane and carbon dioxide produced through the anaerobic digestion of biomass or thermal processes using biomass, including biomass from waste.

Furthermore, they can be classified into three other groups according to the origin of their raw materials. These groups are as follows [27]:

- First-generation biofuels: produced from primary food crops such as corn, sugarcane, palm oil, among others.
- Second-generation biofuels: produced from non-food crops, agricultural residues, the organic part of municipal solid waste, and forest and agricultural residues.
- Third-generation biofuels: biofuel production routes that are in the early stages of research and development or are significantly distant from commercialization (e.g., biofuels from algae, hydrogen from biomass, etc.).

Among its potential uses, the most promoted is transportation. For instance, by 2030, the EU aims to increase the share of renewable energy in transport to at least 14%, including a minimum share of 3.5% for advanced biofuels. Their incorporation aims to mitigate climate change, diversify energy sources, and secure the energy supply [27,28].

Currently, the most widely incorporated and utilized biofuels are bioethanol and biodiesel. As mentioned earlier, there are multiple sources that can be used for their production. However, most of these current biofuels are first-generation biofuels. In the case of bioethanol, its primary raw materials, accounting for approximately 94% of bioethanol produced in the EU, are wheat, corn, and sugar (in the EU's case, derived from sugar beet). On the other hand, for biodiesel, the main feedstocks are rapeseed oil, used cooking oil (UCO), and palm oil, which account for around 84% of the EU's biodiesel production. (The mentioned percentages result from the selection and combination of the three main raw materials for each biofuel [9,11]).

Regarding the raw materials for bioethanol production in the EU, these are widely covered by internal production, with wheat being entirely procured domestically and 95% of sugar beet and 80% of corn also originating within the EU. Due to this scenario, only 20% of corn imported to the EU was considered for the analysis of the social and environmental impacts of bioethanol's imported raw materials (see Table 11). On the other hand, in the case of biodiesel, except for rapeseed oil (95% produced in the EU), the production heavily relies on imported raw materials, with import percentages reaching 65% for UCO and 100% for palm oil.

**Table 11.** Biofuels sector—exported raw materials' origin and SPI values.

| | Raw Material | Top Three Non-EU Exporters [10] | Export Share [10] | $EV_x$ [3] | Averaged SPI per Raw Material | I%/P% | $IV_m$ | $PV_m$ | $CV_m$ | MS% | $SV_s$ |
|---|---|---|---|---|---|---|---|---|---|---|---|
| Bioethanol | Sugar beet | - | - | | | 0%/100% | 0 | 85.29 | 85.29 | 2% | |
| | Wheat | Canada | 63% | 88.17 | 84.96 | 4%/96% | 3.11 | 82.17 | 85.28 | 4% | |
| | | Russia | 20% | 71.99 | | | | | | | |
| | | Australia | 17% | 87.83 | | | | | | | |
| | Corn | USA | 53% | 84.65 | 81.01 | 19%/81% | 15.40 | 69.08 | 84.48 | 12% | |
| | | Argentina | 28% | 78.64 | | | | | | | |
| | | Ukraine | 19% | 74.17 | | | | | | | |
| Biodiesel | Rapeseed oil | Canada | 73% | 88.17 | 83.83 | 5%/95% | 3.85 | 81.37 | 85.22 | 40% | 79.86 |
| | | Russia | 19% | 71.99 | | | | | | | |
| | | Belarus | 8% | 71.49 | | | | | | | |
| | UCO | China | 52% | 65.74 | 70.59 | 65%/35% | 46.19 | 29.47 | 75.67 | 23% | |
| | | Indonesia + Malaysia | 29% | 70.375 | | | | | | | |
| | | US | 18% | 84.65 | | | | | | | |
| | Palm oil | Indonesia | 64% | 66.67 | 69.06 | 100%/0% | 69.06 | - | 69.06 | 19% | |
| | | Malaysia | 34% | 74.08 | | | | | | | |
| | | Guatemala | 2% | 60.21 | | | | | | | |

Similar to the approach used for the paper sector, a conversion factor was used to calculate the MS% of each material. In the case of liquid biofuels, the volumes produced of each material were converted into tonnes of oil equivalent (TOE), considering that each liter of bioethanol is equivalent to $5.05 \times 10^{-4}$ TOE and each liter of biodiesel to $7.92 \times 10^{-4}$ TOE. Thus, with comparable values despite the different products, the methodology can be applied, and the main data and results are shown in Table 11.

The EU liquid biofuels sector is estimated to have an SV of 79.86, considering the main raw materials used and their origin. This represents a significant reduction compared to the EU SPI. It is also noteworthy to mention that the production of biofuels is accompanied by certain potential drawbacks, such as significant water and energy consumption, the release of greenhouse gases from intensive agriculture, the loss of biodiversity and natural habitats, and competition with food production [29]. It is precisely because of these factors that the transition to other generations of biofuels is of paramount importance.

The biofuel sector is a significant job generator within the EU, as evidenced in Table 12, with over 26,000 jobs created (59% attributed to biodiesel and the remaining 41% to bioethanol). Additionally, it contributes to a turnover of 15.1 billion euros, translating to more than 569,000 euros per employee.

**Table 12.** Socioeconomic data of the main liquid biofuels in the sector. Data source: [11].

|  | Employment, 2020 | Turnover (Million EUR), 2020 | Sectoral Bio-Based Output Share, 2020 | Turnover (EUR) per Person Employed |
|---|---|---|---|---|
| Bioethanol | 10,836.18 | 7214.68 | 5.20% | 665,795.51 |
| Biodiesel | 15,742.61 | 7908.81 | 12.50% | 502,382.39 |
| Total | 26,578.79 (sum) | 15,123.49 (sum) | - | 569,005.96 (average) |

### 3.6. Bio-Based Electricity

In the field of bioenergy, along with the biofuels previously discussed and bioheat, there is bio-based electricity. Bio-based electricity production reached over 116 TWh in 2022, which represents 4.4% of the total energy produced in the EU, and 11.17% of that came from renewable sources [30]. The EU's consumption of such energy is almost entirely covered by its own production, i.e., 96%. For this reason, imports of this type of energy are not considered throughout the analysis of this sector.

To obtain bio-based electricity, biofuels and waste are consumed, which, according to the SIEC codification, are classified into several large groups listed in Table 13.

Around half of the EU's bio-based electricity is produced from solid fuels (51%), followed by biogases, which account for 34%, municipal renewable waste with 12%, and the remaining 4% from bioliquids. In turn, each of the first two divisions, the most representatives of the sector, are mainly made up of a single source. In the case of solid fuels, these are "Other wood fuels, wood residues and co-products" (5119) and, in the case of biogases, "Other biogases from anaerobic fermentation" (5320). As previously stated, both are mostly produced within the EU, resulting in the absence of external influences on their SPI. Therefore, the EU SPI of 85.29 directly applies to the bio-based electricity sector. The socioeconomic impacts of this sector (NACE code: D3511) are shown in Table 14.

**Table 13.** SIEC classification of biofuels and waste. Adapted from [31].

| Section | Division | Group | Class | Product |
|---|---|---|---|---|
| 5 |  |  |  | Biofuels |
|  | 51 |  |  | Solid biofuels |
|  |  | 511 |  | Fuelwood, wood residues, and by-products |
|  |  |  | 5111 | Wood pellets |
|  |  |  | 5119 | Other fuelwood, wood residues, and by-products |
|  |  | 512 | 5120 | Bagasse |

**Table 13.** *Cont.*

| Section | Division | Group | Class | Product |
|---|---|---|---|---|
| | | 513 | 5130 | Animal waste |
| | | 514 | 5140 | Black liquor |
| | | 515 | 5150 | Other vegetal material and residues |
| | | 516 | 5160 | Charcoal |
| | 52 | | | Liquid biofuels |
| | | 521 | 5210 | Bio-gasoline |
| | | 522 | 5220 | Bio-diesels |
| | | 523 | 5230 | Bio-jet kerosene |
| | | 529 | 5290 | Other liquid biofuels |
| | 53 | | | Biogases |
| | | 531 | | Biogases from anaerobic fermentation |
| | | | 5311 | Landfill gas |
| | | | 5312 | Sewage sludge gas |
| | | | 5319 | Other biogases from anaerobic fermentation |
| | | 532 | 5320 | Biogases from thermal processes |
| 6 | | | | Waste |
| | 61 | 610 | 6100 | Industrial waste |
| | 62 | 620 | 6200 | Municipal waste |

**Table 14.** Socioeconomic data of the bio-based fraction of the bio-based electricity sector. Data source: [6].

| | Employment, 2020 | Turnover (Million EUR), 2020 | Sectoral Bio-Based Output Share, 2020 | Turnover (EUR) per Person Employed |
|---|---|---|---|---|
| Bio-based electricity | 36,716 (sum) | 690,629 (average) | - | 215,565 (average) |

## 4. Potential for Bio-Based Products from "Secondary Raw Materials"

In the previous section, each main sector of the European bioeconomy was described in terms of its social, environmental, and economic aspects. This analysis revealed that many of the sectors that make up the EU bio-based economy are highly dependent on primary raw materials. This scenario diverges from current EU strategies that promote circularity, resource efficiency, and the valorization of residual and byproduct streams. Nonetheless, numerous initiatives have been launched to develop and enable more sustainable value chains, largely supported by said strategies and regulatory evolution. While bottlenecks exist in terms of technical maturity for some processes, complex supply chains and feedstock instability, several potential routes for waste streams are envisioned.

This section presents the description and generation of main biogenic wastes in the EU, followed by the results of a literature review aimed at identifying the different types of waste that are or could be used as raw materials for the production of various bio-based products. A series of recommendations is then proposed, aimed at avoiding or trying to reduce competition between sectors of the EU bio-based economy.

### 4.1. Biowaste Suitable as Raw Material

Using the NACE categories and counting only the volumes of non-hazardous waste (the easiest to reintegrate in a value chain) generated by the EU, seven waste categories can be distinguished, which would encompass most of the biowaste. As shown in Table 15, more than 70% of this biowaste is made up of three categories, i.e., vegetal waste, wood waste, and paper and cardboard waste.

**Table 15.** EU-27 non-hazardous biogenic waste categories and generated amounts (2020). Data source: [32].

| Non-Hazardous Biogenic Waste | Description | Generation (Tons) | % of Total | Cumulative % |
|---|---|---|---|---|
| Vegetal wastes | – Vegetal waste from food preparation and products, including sludges from washing and cleaning. | 53,640,000 | 27% | 27% |
| Wood wastes | – Wooden packaging.<br>– Sawdust, shavings, and cuttings.<br>– Waste bark, cork, and wood from the production of pulp and paper.<br>– Wood from the construction and demolition of buildings.<br>– Separately collected wood waste. | 46,400,000 | 23% | 50% |
| Paper and cardboard wastes | – Paper and cardboard general waste.<br>– Paper and cardboard waste from sorting and separate collection. | 43,490,000 | 22% | 71% |
| Animal and mixed food waste | – Animal waste from food preparation and products, including sludges from washing and cleaning.<br>– Mixed wastes from food preparation and products, including biodegradable kitchen/canteen wastes, and edible oils and fats. | 25,240,000 | 12% | 84% |
| Common sludges | – Wastewater treatment sludges from municipal sewerage water and organic sludges from food preparation and processing. | 17,270,000 | 9% | 92% |
| Animal feces, urine, and manure | – Slurry and manure, including spoiled straw. | 13,950,000 | 7% | 99% |
| Textile wastes | – Textile and leather waste.<br>– Textile packaging.<br>– Worn clothes and used textiles.<br>– Waste from fiber preparation and processing (man-made and natural fibers).<br>– Waste tanned leather.<br>– Separately collected textile and leather waste. | 1,950,000 | 1% | 100% |
| Total | | 201,940,000 | | |

Following the NACE classification of activities, as illustrated in Figure 4, it can be observed that more than half of this waste comes from households (29%) and industry/manufacturing (27%), while the rest is mainly made up of water supply, sewerage, waste management, and remediation activities (16%), services (except the wholesale trade of waste and scrap metal; 13%), and agriculture, forestry, and fishing (9%).

Furthermore, it can be observed that the types of waste generated in each activity vary substantially, and cases can be observed in which a single activity is the main source of a specific type of waste. This occurs with "Agriculture, forestry, and fishing" generating most "Animal faeces, urine, and manure", and with "Water supply; sewerage, waste management, and remediation activities" generating most "Common sludges".

The end of life of these waste categories (2020) is depicted in Figure 5, showing that recycling and energy recovery cover the largest portion of the waste streams. Important to mention is the fact that there are no specific data available regarding the recovery via recycling, i.e., to which value chains and products the waste streams are directed. It is likely that most routes involve the use of organic waste as compost and animal feed.

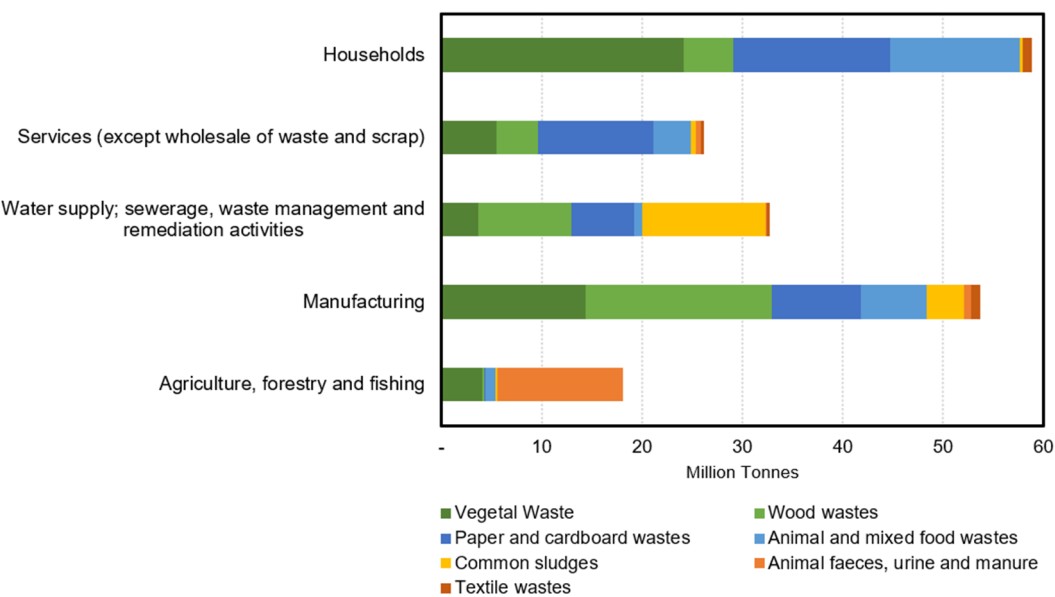

**Figure 4.** Waste generation per activity, year 2020. Data source: [32].

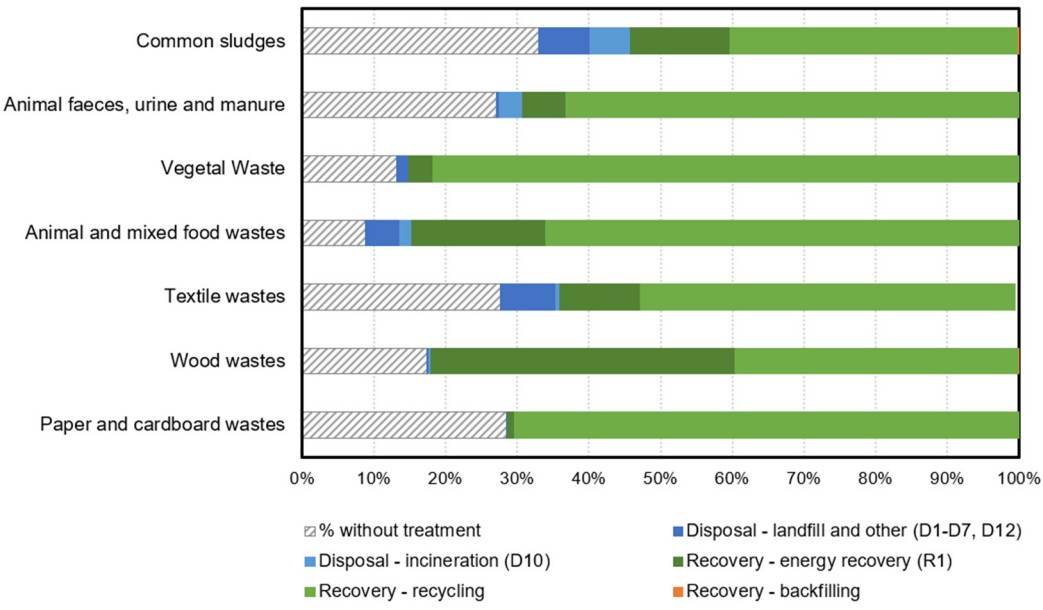

**Figure 5.** Share of treatments used in each bio-residue. Data source: [33].

However, as also depicted in Figure 5, all the waste types have fractions that are not being treated, with percentages of around 30% of the total generated in the case of paper waste, textile waste, animal feces, urine and manure, and common sludges. This represents ca. 39.86 million tonnes of untreated organic waste being uncontrollably disposed of in the environment, leading to pollution and the loss of potential feedstocks for the bio-based economy and/or the generation of renewable energy.

### 4.2. Potential of These Alternative Raw Materials in the EU Bioeconomy

So far, not all sectors of the bio-based economy are able to use all the waste streams described above. Through a bibliographic consultation, routes were identified for converting said wastes into raw materials for a new range of bio-based products. The results were compiled in Table 16, where, for each waste type described in Section 4.1, a set of possible secondary raw materials and applications is identified and referenced.

**Table 16.** Potential applications for the different types of biowaste in the bio-based economy.

| Type of Waste | Raw Materials | Bio-Based Sector and Applications |
| --- | --- | --- |
| Vegetal wastes | – Agriculture/horticulture (corn cob, corn husk, rice husk, rice straw, wheat bran, and agricultural residues).<br>– Fruit, vegetables, cereals, and edible oils (peel waste, seed waste, and fruit waste).<br>– Sugar processing (sugarcane bagasse).<br>– Dairy products (cafeteria waste and municipal solid wastes).<br>– Baking waste.<br>– Alcoholic and non-alcoholic beverages' waste. | Bio-based chemicals, pharmaceuticals, rubber, and plastics [34–48]<br>Biomaterials, enzymes, pharmaceuticals, cosmetic ingredients, sugars, natural pigments, biofertilizers, dairy products, food supplements, bioactive compounds, surfactants, cellulosic products, biocompatible compounds, and materials<br>Textile and textile products [45,48–52]<br>Fibers/fabrics, ropes, and overall textiles used in the food industry, biomedical applications, and machinery<br>Paper [45,46,48,51–55]<br>Bacterial cellulose, pulp from biowaste for paper production, and nanocellulose<br>Liquid biofuels/energy [42–44,47,56,57]<br>Biogas, bioethanol, biodiesel, hydrogen, bio-oil, biomethane, syngas, and electricity |
| Wood wastes | Sawdust, lignocellulosic residues, bark, wood chips, wood pulp, lumber wastes, plywood waste, particle board, timber, wood and garden waste, and glued or painted wood. | Bio-based chemicals, pharmaceuticals, rubber, and plastics [34,40,42,46,47,58–61]<br>Biomaterials, enzymes, pharmaceuticals, sugars, chemical products, biofertilizers, adsorbents, soil conditioners, nanocellulose, and biochar<br>Textiles and textile products [49,50,62]<br>Fibers/fabrics and leather-like material<br>Paper [46,63]<br>Bacterial cellulose, pulp from biowaste for paper production, and nanocellulose<br>Wood products and furniture [59,64,65]<br>Wood-based materials, furniture, and building<br>Liquid biofuel/energy [34,47,65–68]<br>Bioethanol, bio-oil, biogas, and syngas |
| Paper and cardboard wastes | Wastepaper or cardboard | Bio-based chemicals, pharmaceuticals, rubber, and plastics [35,37,47,69]<br>Biomaterials, nanocellulose, and biochar<br>Textiles and textile products [49,70,71]<br>Fibers/fabrics<br>Paper [34,46]<br>Bacterial cellulose, recycled paper, and nanocellulose<br>Liquid biofuel/energy [34,47,67,68,72]<br>Bioethanol, syngas, and electricity |
| Animal and mixed food waste | – Chitin/chitosan (shrimp shells and crustacean shell waste).<br>– Oil/animal fat (waste cooking oil and fish oil).<br>– Food/kitchen waste/organic fraction of municipal solid wastes.<br>– Collagen/gelatin.<br>– Others (feather quill, blood plasma, lactose, . . .) | Bio-based chemicals, pharmaceuticals, rubber, and plastics [34,35,39,40,43–45,47,69,73–75]<br>Biomaterials, enzymes, pharmaceuticals, cosmetic ingredients, chemical products, lubricants, biofertilizers, bioactive compounds, dyes, bacterial cellulose, and biochar<br>Paper [45,51,52]<br>Bacterial cellulose<br>Textiles and textile products [45,51,52,73]<br>Fibers/fabrics and bacterial cellulose<br>Liquid biofuel/energy [34,43,44,47,73,75–78]<br>Biogas, bioethanol, biodiesel, hydrogen, bio-oil, and biomethane |

**Table 16.** *Cont.*

| Type of Waste | Raw Materials | Bio-Based Sector and Applications |
|---|---|---|
| Common sludges | – Domestic wastewater.<br>– Industrial wastewater (wood mill effluent, oil industry effluent, paper mill sludge, and textile mill/effluent).<br>– Sewage sludge/excess biological sewage sludge (EBSS). | Bio-based chemicals, pharmaceuticals, rubber, and plastics [34,35,42–44,48,64,79–83]<br>Plastics, pharmaceuticals, cosmetics, sugars, biofertilizer/digestate, biopesticides, enzymes, proteins, solvents, and bacterial cellulose<br>Textiles and textile products [48,51,52,81,82]<br>Fibers/fabrics and bacterial cellulose<br>Paper [48,51,52]<br>Bacterial cellulose<br>Liquid biofuel/energy [34,43,44,82]<br>Biogas, bioethanol, biodiesel, dihydrogen, bio-oil, and biomethane |
| Animal feces, urine, and manure | Manure | Bio-based chemicals, pharmaceuticals, rubber, and plastics [34,43,44,47,84]<br>Biomaterials, pharmaceuticals, cosmetic ingredients, biofertilizers, and biochar<br>Liquid biofuel/energy [43,44,47,84]<br>Biogas, bioethanol, biodiesel, hydrogen, bio-oil, and biomethane |
| Textiles | – Pre-consumer waste (ginning waste, yarn manufacturing waste, and fabric manufacturing waste).<br>– Post-consumer garment waste. | Bio-based chemicals, pharmaceuticals, rubber, and plastics [85–88]<br>Fertilizers, biochar, and cellulose acetate<br>Textile and textile products [34]<br>Textile recycling<br>Liquid biofuel/energy [89–91]<br>Syngas, bio-oil, biogas, bio-alcohol, hydrogen, biodiesel, and electricity |

As reflected in Table 16, the diversity of uses for biowaste in the main sectors of the EU bio-based economy is wide. Furthermore, said waste streams can also be used in other sectors/activities, such as construction or animal feed [92–95]. Overall, there is significant potential for improvement in the reintegration of biowaste streams into the bioeconomy. However, for meaningful progress to occur, it is important to address the existing barriers. These barriers cover all sorts of issues, including policy and regulation, stakeholder perceptions, investment challenges, and various impediments, such as those related to intellectual property, human resources, and effective collaboration, in addition to the technical challenges on utilization of biowaste as a feedstock, due to the typical heterogeneity of said streams, unstable and decentralized supply chains, and the lack of mature and/or feasible recycling technologies [96]. In general, the use of secondary raw materials, as will be discussed in more detail in Section 5.2, is surrounded by multiple barriers that end up, among other consequences, increasing costs for companies, reducing the demand for byproduct and/or waste streams, which end up being used at lower hierarchical levels [92]. One notable constraint, which is likely to become more important with time and increasing demand, is the instability of raw material supply. This issue could be exacerbated by competition, as illustrated in Table 17, which can emerge among the evaluated sectors due to the shared use of various waste types as potential raw materials for their products. This is especially the case for less heterogeneous streams that can be converted into high-value products, e.g., dry wood and vegetal wastes and sorted paper/cardboard wastes.

**Table 17.** Comparison between main bio-based sectors and biogenic waste categories that can potentially serve as feedstock.

| Bio-Based Sectors/Biogenic Wastes | Textiles and Textile Products | Wood Products and Furniture | Paper | Bio-Based Chemicals, Pharmaceuticals, Rubber, and Plastics | Bio-Based Electricity |
|---|---|---|---|---|---|
| Animal and mixed food waste | X | - | X | X | X |
| Vegetal wastes | X | - | X | X | X |
| Paper and cardboard wastes | X | - | X | X | X |
| Wood wastes | X | X | X | X | X |
| Common sludges | X | - | X | X | X |
| Animal feces, urine, and manure | - | - | | X | X |
| Textile wastes | X | - | - | X | X |

Considering the great diversity of materials within the waste groups and the fact that a quantitative analysis could not be carried out based on Table 17, the comparison presented can be analyzed from two perspectives: that of the bio-based sectors and that of the waste streams.

From the first perspective, except for the wood products and furniture sector, it is worth highlighting the great diversity of waste categories that bio-based sectors can use as raw materials. For instance, "Chemical, pharmaceutical, rubber and plastics" bio-products and "bio-based electricity" can potentially use all the waste groups studied. These sectors would be followed by the textiles and textile products sector and the paper sector, with the potential to use, respectively, six and five of the seven waste groups.

From the second perspective, the only waste group that can be used by all sectors is wood waste. With this case left aside, animal and mixed food waste, vegetal waste, and paper and cardboard waste are of great relevance, given their potential for use in all sectors except for wood products and furniture, which, as mentioned above, can only use one type of waste. In the case of the rest of the waste, its potential is somewhat lower, as it is used in three sectors.

A cascading scheme could prevent competition, in which sectors with more limited possibilities in terms of raw materials would first absorb the needed volumes, and sectors with more flexibility in processing raw materials could absorb the "less competitive" feedstocks. Important to mention is the fact that a much more in-depth analysis would be needed to fully address the feedstock competition issue, which includes many other parameters, such as regional particularities and the geographic location of wastes and conversion units, market demands for specific products, local industries, etc.

## 5. Discussion

### 5.1. Reflections on the SVs, Employment, and Turnover

With the aim of assessing, from a multidisciplinary perspective, the sectors comprising the bio-based economy, a methodology has been developed, as described in Section 2.1. Throughout Section 3, this methodology was applied to each of the six studied sectors, resulting in a single indicator per sector, SV. This value describes the socio-environmental status of the sector based on the primary raw materials used in the production of its main bio-based products. It considers production and import ratios, the origin of the imported fraction, the Sustainable Process Index (SPI) of the main countries exporting to the EU, their respective export shares, and the percentage of market share for each raw material.

However, the SV is not the sole outcome derived from the application of this methodology. As depicted in Figure 6, it enables the characterization of each analyzed sector based

on its dependence on a single material, its reliance on material imports, the average socio-environmental status of the imported fraction, and the presence or absence of biowaste as a feedstock among its main biogenic raw materials.

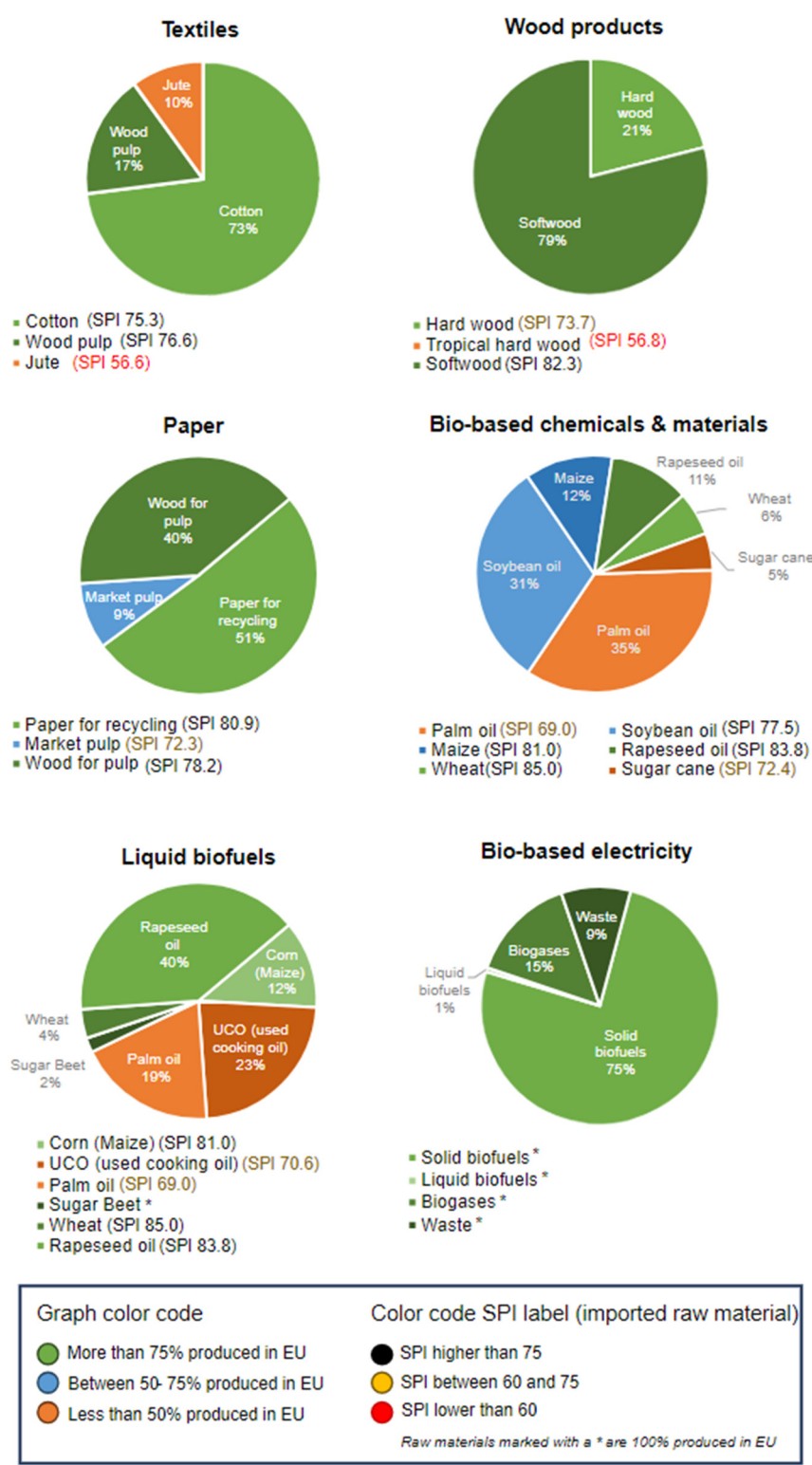

**Figure 6.** Sector overview based on each sector's main raw materials, EU dependency, and SPI.

When focusing only on the SV, the top-ranking sector is bio-based electricity with a score of 85.29, attributed to its absence of raw material imports and 75% of production exclusively sourced from solid biofuels. Following closely is the sector of wood and

furniture products, with a score of 85.17. Similar to the previous sector and others with high SVs, this sector has its raw material demand largely met by EU production. It primarily utilizes softwood, constituting 79% of the consumed bio-based raw materials.

Subsequently, the sector of textiles and textile products, the last with an SV exceeding 80, is noted with a score of 80.66. Despite having a lower SV than the preceding sectors, it is the first to include waste (recycled paper) among its main raw materials. In comparison to the previous sectors, it exhibits lower dependence on a single material, and its current demand is mostly satisfied by EU production.

In the case of liquid biofuels, which display more variety in their primary raw materials, including waste (UCO), the SV is 79.86. This is the result of higher dependence on imports, especially for a couple of products, UCO and palm oil, which, in turn, exhibit a considerably low average SPI.

Lastly, biochemicals have an SV of 78.18, akin to biofuels, primarily due to the necessity of importing many raw materials to meet sector demand. In some cases, these imports originate from countries with an SPI considerably lower than the EU average.

In all cases, and particularly the ones where the SV is lower, there are multiple pathways to explore, which are listed below:

- Increase the production of raw materials within the EU.
- Substitute raw material suppliers with countries with a better SPI or foster actions to positively impact said countries in terms of socio-environmental aspects (see Figure 3), leading to the increase of their SPI and more fair living and working conditions in the raw material supply chains.
- Increase the proportion (MS%) of raw materials with better socio-environmental indicators.
- Substitute raw materials with lower associated SPI for raw materials with higher SPI values.
- Use biowaste as raw material and promote the development of novel, circular supply chains based on secondary raw materials, rather than virgin ones.
- For the particular case of UCO, favor the internally produced streams of this type of biowaste, which currently rely on imports of more than 50%.

Besides the environmental and social perspectives, Section 3 also presents socioeconomic data in the form of employment and turnover generation, whose share of the overall bio-based economy is shown in Table 18.

**Table 18.** Socioeconomic contributions.

| Bio-Based Sector | Employment, Shares % * | Turnover, Shares % * | Bio-Based Product Output Share ** |
|---|---|---|---|
| Textiles and textile products | 23% | 10% | 42.20% |
| Wood products and furniture | 41% | 25% | 72.40% |
| Paper | 19% | 26% | 99.50% |
| Bio-based chemicals, pharmaceuticals, rubber, and plastics | 15% | 33% | 19.50% |
| Liquid biofuels | 1% | 2% | 93% |
| Bio-based electricity | 1% | 4% | 5.80% |

* Share percentage based on the total bio-based economy. ** Average percentage of bio-based matter in products labeled as "bio-based" within each sector.

The two first parameters describe the status of the EU bio-based economy, where employment shares are dominated by the sectors "Wood products and furniture", "Textile and textile products", and "Paper", respectively. This is expected since these are traditional and well-developed sectors. Notably, while the textiles sector covers a significant employment share, its turnover is relatively low, evidencing a lower value of said products compared to wood- and paper-derived products. This is likely related to the higher presence of cheap raw materials from countries outside the EU and to a much more de-

centralized and multistep value chain that involves several actors. On the contrary, the sector "Bio-based chemicals, pharmaceuticals, rubber, and plastics" is still in an early development stage for many bio-based products, showing a rather low employment share, considering how diverse it is in terms of products. Nonetheless, its turnover shares are the highest of all sectors, which is a result of the significant value added of most bio-based chemicals, biomaterials, and bioactive compounds. Finally, the sectors "Liquid biofuels" and "Bio-based electricity" show both low employment and turnover shares. While the liquid fuel markets are huge, they are largely dominated by fossil-based value chains. In the case of electricity, other renewable sources are growing steadily in the EU (e.g., solar, wind, and hydroelectric). Thus, the bio-based fuels and electricity markets are still small compared to other bio-based sectors that rely much more on biogenic feedstocks, such as wood products and paper. Furthermore, the prices of fuel and electricity are low compared to other bio-based products.

Another important parameter considered is the "Bio-based product output share", which estimates the bio-based percentage in the products labeled as bio-based in each sector. As shown in Table 18, this parameter varies significantly, depending on the sector, indicating where the real bio-based content of the product is very high (i.e., in paper products and in the two considered liquid biofuels, ethanol and biodiesel) and, most importantly, where this content can be increased. For instance, textile and wooden products (e.g., furniture) tend to be multimaterial and include coatings, additives, dyes, adhesives, fibers, and hard pieces made of petro-based feedstocks, metals, and inorganic materials. This not only lowers the bio-based content but also complicates the recycling and sustainability profile of products, as many elements are often non-biodegradable, are difficult to sort, and have safety concerns. In the hybrid sector of electricity, mixes often have low bio-based content—but, fortunately, an increasing renewable energy content when considering the above-mentioned sustainable sources. And finally, the "Bio-based chemicals, pharmaceuticals, rubber, and plastics" sector has a lot of potential to expand on the bio-based content of its products, averaging less than 20 wt% currently. For instance, the bio-based content of chemicals labeled as "bio-based" is typically lower than 10 wt%, the rubber market is mostly dominated by synthetic rubber (petro-based), and the share of bioplastics in the EU is still very low (ca. 1 wt% [97,98]). Considering the added value of these bio-based products, the development of this sector could bring significant positive impacts to the EU bio-based economy. This overview also highlights pathways for increasing the sustainability of each sector, which are not necessarily the same: while some sectors with a high bio-based share can benefit from the reintegration of second raw materials (i.e., biowaste streams) in their value chains, other sectors can benefit from both an increase in the bio-based content of their products and the reintegration of secondary raw materials as feedstocks. This discussion will be further expanded in Section 5.2.

Based on the parameters set out in these reflections, the following descending rankings were established for the sectors that make up the EU bio-based economy. The bio-based electricity sector was not considered in the rankings since it is a relatively small sector with particularities and difficult raw material traceability, evidenced by the very low reported "bio-based share output" due to the energy mixes that vary substantially geographically. It is also a sector greatly oriented towards decarbonization via renewable energy sources such as solar, wind-powered, etc.

1.  Paper: This sector has a high SV, as well as good employment and turnover figures, all this while presenting the highest bio-based product output.
2.  Wood products and furniture: As mentioned above, the wood products and furniture sector has the highest SV of all sectors, as well as very good figures in the socioeconomic parameters studied; however, its bio-based output share can be further increased.
3.  Textiles and textile products: Although this sector does not show poor results in terms of socioeconomic parameters, there is room for improvement in SV (impacted by a rather high raw material dependency), especially in the bio-based output share.

4. Bio-based chemicals, pharmaceuticals, rubber, and plastics: Although, in socioeconomic terms, the sector shows very good figures, especially in terms of turnover, its SV and bio-based output share are rather low. This evidences a high dependency on the importation of raw materials, low circularity, and the dominance of fossil-based building blocks in most chemical formulations and production processes.

5. Liquid biofuels: This sector has a very high bio-based product output share, but its SV and final contribution to the generation of employment and turnover in the EU is currently very low compared to the other sectors. Important to mention is the fact that the liquid transportation fuels sector is going through a transition in which electrification plays a major role in urban mobility and the biggest push for biofuels' developments targets sectors that are difficult to decarbonize, such as aviation, heavy road transport (trucks), and marine transportation.

The results of both socio-environmental and socioeconomic analyses are summarized in Figure 7, where, for each of the sectors, the obtained SV is shown, as well as its contribution in terms of employed individuals and turnover generated in relation to the overall EU bio-based economy.

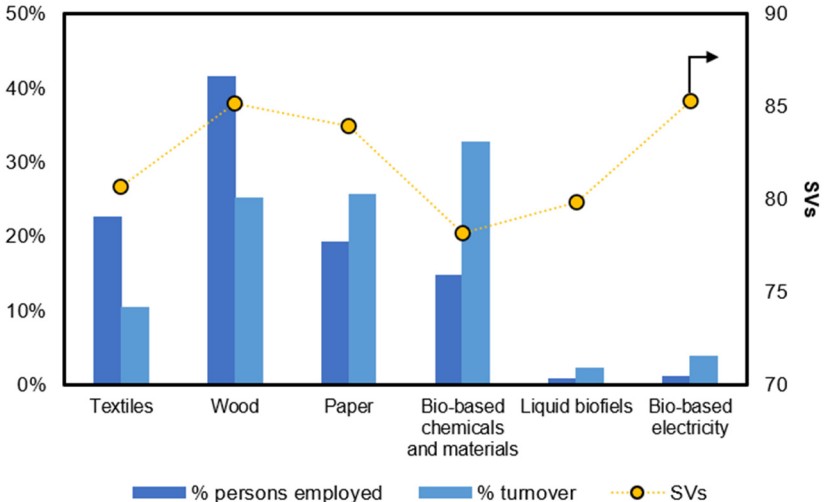

**Figure 7.** Overview of the socio-environmental and socioeconomic indicators per bio-based sector in the EU.

### 5.2. Reflections on the Use of Biowaste as a Raw Material

As previously mentioned, the bioeconomy, including the bio-based economy, faces multiple barriers to its development. The most significant barriers include the following: (i) policy constraints on the demand side, notably the insufficient promotion of bio-based products (such as incentives, taxation, market support, standards, and product specifications); (ii) perception barriers among stakeholders, stemming from limited awareness and communication regarding the benefits of bio-based products; and (iii) investment obstacles arising from factors like the absence of tangible products, an extended return on investment periods, and regulatory complexities [96].

While issues related to raw materials are not explicitly mentioned among these main obstacles, as the demand for bio-based raw materials increases, either through companies' own initiatives or as a result of various EU strategies promoting circularity and resource efficiency, it is foreseeable that challenges may arise regarding the supply chain for bio-based raw materials. This is particularly evident given the clear interest in promoting the valorization of biowaste, which, as observed in Section 4.2, has multiple potential applications in the bioeconomy. However, there are several regulatory, technical, and social barriers that constrain or complicate their use.

Within the regulatory and legal domain, various issues stand out as barriers to the valorization of biowaste into products such as animal feed or fertilizers. These issues,

stemming from reasons such as a significant lack of flexibility, unrealistic purity demands, or the rejection of certain biowaste fractions, hinder or outright prevent the effective utilization of biowaste. Furthermore, the implemented legislation is considered to be unclear and complex, featuring numerous documentation systems and a lack of policy amortization, leading to inconsistent applications and interpretations. Another notable issue is the lack of transparency in market rules, remuneration agreements, and market support mechanisms, whether they involve regulatory requirements or incentives for utilizing biowaste as a resource, subsidies, or technical support [92,93,99].

Among the technological barriers, economic factors play a significant role, such as the challenge of ensuring that the final products are competitively priced. Additionally, uncertainties related to supply, as well as the volatility of raw materials in terms of cost, logistics, and quality, contribute to the obstacles faced in this context. However, not everything in this domain is confined to economic aspects. Within this realm, numerous challenges persist that demand further technological development. These include addressing the heterogeneity of biowaste, optimizing certain processes, and enhancing their properties to be fit for conversion into products of higher value [93,99].

Within the social sphere, a notable characteristic is the lack of awareness and information, which can be responsible for the misconception surrounding the valorization of biowaste and biologically based products among consumers, businesses, and organizations. This lack of awareness may lead to the avoidance of such products due to misguided ideas [92,99].

The existence of barriers is well illustrated by the scarcity of biowaste among the main raw materials used by each sector, the basis of the analysis in Section 3, which shows that the EU bio-based sectors mainly use primary sources for the manufacture of their products.

Despite these barriers and the current widespread underutilization of biowaste for the production of bio-based products, a general increase in demand is expected in the coming years due to strong policies favoring circular and sustainable value chains and ever-growing environmental concerns. This could give rise to competition issues among sectors comprising the bio-based economy, each of which competes for different biowaste resources, namely Animal and mixed food waste, Vegetal wastes, paper and cardboard wastes, wood wastes, common sludges, animal feces, urine and manure, and textile wastes. With this in mind, the analysis conducted in Section 4 has provided insights into the versatility of each sector in utilizing bio-based residues and by-products, as well as the potential for these waste materials to be employed in a wide range of applications. As a result of this analysis, it has been observed that the bio-based chemicals, pharmaceuticals, rubber, and plastics and bio-based electricity sectors have the highest number of waste groups' possibilities to be used as raw materials. Indeed, in the case of bio-based electricity, there is great flexibility in using either solid, gas, and liquid biowastes as sources (vide supra, Table 13). On the other hand, in the field of bio-based chemicals, pharmaceuticals, rubber, and plastics, while a specific product may not be producible from all types of biowaste, the diversity of products in this sector potentially allows the use of all biowaste types as feedstock.

The textiles and textile products sector also shows flexibility in terms of using biowaste streams as feedstock. Nevertheless, it is noteworthy to emphasize that the ability to utilize this diversity of waste is almost exclusively attributed to the possibility of obtaining bacterial cellulose and its properties, which enables the use of various waste materials as raw materials for the production of fibers. Finally, in the wood and furniture products sector, the waste integration options are limited to wood as a result of the nature of the sector. Other important products associated with wood products, but belonging to other sectors, such as adhesives, could be made from other types of biowaste.

When looking at it from the opposite perspective, it is evident that wood waste offers the widest range of applications, being versatile enough to be used across all sectors of the bio-based economy. Following closely are animal waste and combinations of food, plant,

paper, and cardboard waste, which find utility in every sector except for wood and furniture products. Other waste categories show potential applications in up to three sectors.

It is worth noting that the scope of the results of this analysis is limited to identifying the existence of means to generate a product within the corresponding sector from some of the studied biowastes. This implies that the differences in technology readiness levels (TRLs) between technologies and processes have not been considered. As mentioned earlier, the need for further technological developments is one of the barriers faced by the bioeconomy, particularly the efficient and feasible conversion of biowastes into valuable bio-based products. In this regard, while certain processes exhibit industrial applicability (TRL 7–9), many processes are still at the pilot stage (TRL 4–6) or even at the laboratory level (TRL 1–3) [100–102].

Currently, the most widely employed technologies in this field are composting and anaerobic digestion. Biowastes originating from food processing and agriculture are the most utilized, as these biowaste streams are better defined and cleaner. Despite having these treatments, a significant fraction of biowastes still ends up in mixed waste streams destined for landfills or incineration. This occurs even in countries with well-established collection systems. Various technologies, such as fermentation, extraction, biodigestion, pyrolysis, gasification, hydrothermal liquefaction, and animal feed production, among others, have the potential to address or alleviate this issue [67,93]. However, they currently face challenges in making the leap to the industrial level due to the need for further development to address issues such as high processing costs, low substrate conversion efficiency, quality improvement, high energy demand, poor technology transfer from academia to industrial application, and the resolution of certain legislative issues [67,93,96]. Although said needs apply to all waste types, according to the literature, those that seem to require more effort to achieve industrial application are sludges [64,81,82]. And finally, despite the aforementioned challenges, several promising technologies are being developed to transform waste streams in secondary raw materials targeting higher-value products. Relevant examples that are reaching or have already reached a commercial stage include the following: (i) the sustainable plastic PEF (polyethylene furanoate) from plant-based sugars [103]; (ii) 100% recycled fibers from textile wastes [104]; (iii) bio-based thermoplastics from heterogeneous waste streams [105]; (iv) high-purity chemical building blocks from lignocellulosic wastes [106,107]; and (v) second-generation ethanol from vegetable wastes (notably sugarcane bagasse and straw) [108]. Based on the results and the future competition that could take place between sectors as the TRL increases, the following recommendations are made to avoid or reduce competition between sectors when incorporating biowaste as a raw material in value chains:

- Prioritize the reintegration of biowastes generated within the sector responsible for their production. For instance, whenever feasible, the textile sector should make use of its own generated textile waste and common sludges.
- Prioritize the utilization of wood waste in the wood and furniture products sector, given the limited variety of waste types that the sector can effectively employ.
- Prioritize the incorporation of biowaste as raw materials in sectors with a lower bio-based product output share and SV, such as the bio-based chemicals, pharmaceuticals, rubber, and plastics or textiles and textile products sectors.

## 6. Conclusions

This study conducted a series of analyses to characterize and compare different industrial sectors that make up the bio-based economy in the EU. Each of these six main sectors has been evaluated according to the socio-environmental situation of the main countries that export raw materials to the EU. Based on this analysis, the bio-based electricity, wood products and furniture, and paper sectors show socio-environmental indicators (represented by the SV) similar to the EU average, since these are sectors in which raw materials are mostly produced internally. On the other hand, the other sectors analyzed (textiles, bio-based chemicals and materials, and liquid biofuels) show lower SVs, given

the greater dependence on imports of the raw materials used and the overall lower socio-environmental indicators (represented by the SPI) of the main exporting countries. This analysis has also resulted in formulating a ranking that presents the sector in descending order based on the socio-environmental and socioeconomic conditions:

1. Paper.
2. Wood products and furniture.
3. Textiles and textile products.
4. Bio-based chemicals, pharmaceuticals, rubber, and plastics.
5. Liquid biofuels.

This overview is complemented by a thorough analysis of the main streams of non-hazardous biowastes generated in the EU and their potential valorization routes according to the most recent literature. Based on the identification of potential pathways for the development of the EU bio-based economy targeting secondary raw materials and the overview of the main bio-based sectors and products, the following recommendations have been formulated to mitigate or alleviate potential future competition that may arise among the sectors, as well as to assist those sectors with lower sustainability indicators (SVs) in improving their socio-environmental scenario:

- Boost the production of the main raw materials in the EU or replace imported raw materials with others produced in the EU.
- Substitute current raw material suppliers with others from countries with better socio-environmental practices (translated from a higher SPI).
- Raise the share of raw materials with a higher SPI and/or replace lower-SPI raw materials with better alternatives.
- Utilize biowaste generated in the EU as raw materials in circular value chains.
- Emphasize the reintegration of biowaste within the sector responsible for its production.
- Prioritize the use of wood waste in the wood and furniture products sector.
- Prioritize the incorporation of biowaste as raw materials in sectors with a lower bio-based product output and poorer socio-environmental indicators (as represented by the SV in this study).
- Actively promote internal (EU) and global actions to improve the social and environmental aspects that impact the social progress index (SPI) of countries. This translates into better working, living, and environmental conditions to produce the needed raw materials.

The most significant barriers that need to be addressed in order to boost the development of the bio-based economy are summarized below:

- Demand-side policy barriers, in particular the lack of development in the promotion of bio-based products (incentives, taxation, market support, standards and product specifications, etc.).
- Stakeholder perception barriers as a result of the lack of stakeholder awareness of bio-based products, as well as a lack of communication about the benefits of bio-based products.
- Investment barriers due to factors such as the absence of visible tangible products, successful product outcomes, and lengthy return on investment periods, as well as various regulatory barriers.

It is important to note that the methodology developed, presented, and applied in this article to assess the different sectors comprising the bio-based economy still has potential for improvement and greater precision. For instance, additional aspects, indicators, or even phases of the life cycle could be incorporated. The inclusion of the latter would shift the analysis away from being solely focused on raw materials, allowing for the evaluation of stages that may have significant socioeconomic and environmental impacts, such as manufacturing. However, at the moment, there are certain limitations due to data availability, reliability, and the differences/particularities of each sector that complicate a direct comparison. Among the most noteworthy aspects that have been detected throughout

the data collection and their subsequent processing are the following: nomenclatures and databases that are not standardized, the lack of information depending on the sector, little real industrial data available to the general audience, and the lack of specific information regarding the end of life of biowaste streams (e.g., the main products obtained from current recycling activities).

**Author Contributions:** Conceptualization, V.F.O. and M.B.F.; methodology, V.F.O.; validation, M.B.F., C.B. and S.Z.; formal analysis, M.B.F.; investigation, V.F.O.; data curation, V.F.O.; writing—original draft preparation, V.F.O.; writing—review and editing, V.F.O., M.B.F. and C.B. All authors have read and agreed to the published version of the manuscript.

**Funding:** The research leading to these results has received funding from the European Union's Horizon Europe Coordination and Support Actions programme under grant agreement No 101059785, project name: SUSTCERT4BIOBASED. This project was funded by the European Union. The views and opinions expressed are those of the authors only, and do not necessarily reflect those of the European Union or the European Climate, Infrastructure, and Environment Executive Agency. Neither the European Union nor the granting authority can be held responsible for them.

**Institutional Review Board Statement:** Not applicable.

**Informed Consent Statement:** Not applicable.

**Data Availability Statement:** Data are contained within the article.

**Conflicts of Interest:** The authors declare no conflict of interest.

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
