# Peer review of "Assessment of EU Bio-Based Economy Sectors Based on Environmental, Socioeconomic, and Technical Indicators"

_sustainability, doi:10.3390/su16051971_

Round 1

Reviewer 1 Report

Comments and Suggestions for Authors

Dear authors

Thank you for your submission that deals with an interesting and very topical subject.

The paper needs some revision in terms of grammatical issues (which can easily be addressed with a copy editor) and editor structure (i.e font in the last line of table1, notes to table are put in different point for example in table 2 and table 18, line 533 double “,” line 702 reference to table 19 perhaps it is 18

The paper needs to be strengthened with a series of text revision measures. It contains many information and for a better comprehension of whole article it is necessary to rewrite some parts.

2 Methodology section needs to better explain the single step.

·       Before the paragraph 2.1 it is necessary to describe the overall methodology, also with a chart that help the reader.

·       Both in Paragraph 2.1 and paragraph 2.2 your speech about literature review but you don’t explain how you did it.

·       Some issue has to be explained or need references: line 132 three main countries, line 158 SPI score range, line 167 criteria of non-hazardous waste generation

3 Analysis of the current situation:

·       please in table 2,4,7,8,11 put the column in the same order o the explanation in methodology,

·       table 3,5,8,10,12,14-line total in some are average in other is not specified. Please check it

4 Potential for bio-based:It is not clear the construction of table 17 in which I suggest to put the sector in the same order of the rest of the article

Comments on the Quality of English Language

Minor editing of English language required

Author Response

Dear Reviewer,

You will find in this attachment your recommendations, followed by the corresponding measures which have been carried out in the paper.

Thank you very much for your contributions to the improvement of the paper.

Please do not hesitate to contact us if any additional information is needed.

Yours sincerely,

Víctor Fernández Ocamica

Reviewer 2 Report

Comments and Suggestions for Authors

The manuscript is very well written and presents a suitable compilation of information on UE Bioeconomy status and future guidelines. In general, dataset has good sources and data analysis is well formulated. I wish to recommend only a few improvements:

1)  Ethanol is bio per se (fully renewable, bio-based), so it is not advisable to use 'bioethanol' as terminology.

2) I miss some discussion on H2 from biomass. Please, considere discussing some relevant aspects because green and blue H2 will certainly play a significant role in renewable transformation of UE and elsewhere. Please, refer to references like https://doi.org/10.3390/su151612641.

Author Response

(The authors gave the same response as above.)

Reviewer 3 Report

Comments and Suggestions for Authors

Sustainability in Bioeconomy and Bioenergy is very important. A new methodology is proposed to link each bio-based sector’s geographical origin, import shares and internal production with socio-environmental impacts based on official databases and indexes. This is the main innovation of this manuscript, which can provide a good evaluation standard reference for other regions. The authors have done a lot of work, especially in organizing the existing research results. However, this manuscript is too long and its article classification is unclear. Suggest reviewing after modification.

The main shortcomings are as follows.

1. The abstract mainly provides the research significance and ideas of this manuscript, but lacks research conclusions.

2. There are too many keywords, which is recommended to be reduced to around 5.

3. The manuscript lacks a literature review.

4. The production of figures and tables is not very standardized in this manuscript, especially Table 11. Suggest modifying the corresponding figures and tables to make them more aesthetically pleasing and scientific.

5. These two paragraphs on line 295-305 are mainly cited from reference [17]. It is recommended to reorganize the relevant text and avoid the phenomenon of repeatedly labeling the same reference.

6. The paragraphs and Table 6 on line 306-316 are mainly cited from reference [5]. They are also like this. There are quite a few similar situations in this manuscript.

7. The writing of some sentences does not quite conform to the language habits of English. Suggest inviting one professional to proofread the manuscript.

8. There is a low-level error. There are two “Figure 5.” on line 636 and 898 in this manuscript.

9. The last figure should not appear in “6. Conclusions”.

10. Is this manuscript a review paper or an empirical paper? I cannot clearly grasp it.

Comments on the Quality of English Language

The writing of some sentences does not quite conform to the language habits of English. Suggest inviting one professional to proofread the manuscript.

Author Response

(The authors gave the same response as above.)

Round 2

Reviewer 1 Report

Comments and Suggestions for Authors

The paper is now well prepared. It could be published

Author Response

Thank you very much for your contributions to the improvement of the Article

Reviewer 3 Report

Comments and Suggestions for Authors

1. There are two titles in Figure 1.

2. The title of Figure 2 cannot be understood.

3. There are low-level errors in this manuscript, such as: [CO2] on line 342.

4. The reference labeling in the tables is very non-standard and inconsistent.

5. The reference [10] has been annotated too many times. I feel that this manuscript is a rewrite of this reference.

6. The title of Figure 6 is too long.

7. The writing style of "6. Conclusions" is very unusual.

8. Is this manuscript a review paper or an empirical paper? I cannot clearly grasp it.

Comments on the Quality of English Language

Minor editing of English language required

Author Response

Dear Reviewer,

You will find in this attachment your recommendations, followed by the corresponding measures which have been carried out in the paper.

Only the question of the conclusions remains unanswered, as we have not been able to see how to improve the paragraph. Could you specify what you are suggesting?

Thank you very much for your contributions to the improvement of the paper.

Please do not hesitate to contact us if any additional information is needed.

Yours sincerely,

Víctor Fernández Ocamica

Round 3

Reviewer 3 Report

Comments and Suggestions for Authors

The standardization of this manuscript is still very poor.

This is already my third time reviewing the manuscript. I hope the authors can take it seriously. The main issues have already been raised before.

Comments on the Quality of English Language

Minor editing of English language required

Author Response

Dear reviewer, 

As you can see in the previous exchanges (compiled in the attachment), we tried our best to accomodate all of your comments, including the more general ones. We have fully reviewed the manuscript another time and a "2nd response" was added whenever applicable in the list of comments from you (see attachment)

We believe we have fully addressed the issues raised, but of course we are open for any constructive and specific feedback to further improve our manuscript. We remind you that the latest version of the manuscript in .doc format, where all changes can be seen with the "track changes" tool, is available in the platform. 

We hope you consider our efforts accordingly and that this is sufficient. 

Best regards,

Víctor Fernández Ocamica
